# Deciphering Bacterial Community Succession and Pathogen Dynamics in ICU Ventilator Circuits Through Full-Length 16S rRNA Sequencing for Mitigating the Risk of Nosocomial Infections

**DOI:** 10.3390/microorganisms13091982

**Published:** 2025-08-25

**Authors:** Hsin-Chi Tsai, Jung-Sheng Chen, Gwo-Jong Hsu, Bashir Hussain, I-Ching Lin, Tsui-Kang Hsu, Jing Han, Shih-Wei Huang, Chin-Chia Wu, Bing-Mu Hsu

**Affiliations:** 1Department of Psychiatry, School of Medicine, Tzu Chi University, No. 701, Section 3, Zhongyang Rd., Hualien 970, Taiwan; cssbmw45@gmail.com; 2Department of Psychiatry, Tzu-Chi General Hospital, No. 701, Section 3, Zhongyang Rd., Hualien 970, Taiwan; 3Department of Medical Research, E-Da Hospital, I-Shou University, No. 1, Section 1, Xuecheng Rd., Kaohsiung 824, Taiwan; 4Division of Infectious Disease, Department of Internal Medicine, Chia-Yi Christian Hospital, No. 539, Zhongxiao Rd., Chiayi 600, Taiwan; 5Department of Earth and Environmental Sciences, National Chung Cheng University, No. 168, Section 1, Daxue Rd., Chiayi 600, Taiwan; 6Department of Family Medicine, Asia University Hospital, No. 222, Fuxin Rd., Taichung 413, Taiwan; 7Department of Kinesiology, Health and Leisure, Chienkuo Technology University, No. 1, Jieshou N Rd., Changhua 500, Taiwan; 8Department of Ophthalmology, Cheng Hsin General Hospital, No. 45, Zhenxing St., Taipei 112, Taiwan; 9School of Medicine, National Yang Ming Chiao Tung University, No. 1001, Daxue Rd., Hsinchu 300, Taiwan; 10Department of Rehabilitation, Ditmanson Medical Foundation, Chia-Yi Christian Hospital, No. 539, Zhongxiao Rd., Chiayi 600, Taiwan; 11Center for Environmental Toxin and Emerging Contaminant, Cheng Shiu University, No. 840, Chengqing Rd., Kaohsiung 824, Taiwan; 12Institute of Environmental Toxin and Emerging Contaminant, Cheng Shiu University, No. 840, Chengqing Rd., Kaohsiung 824, Taiwan; 13Division of Colorectal Surgery, Dalin Tzu Chi Hospital, Buddhist Tzu Chi Medical Foundation, No. 2, Minsheng Rd., Chiayi 600, Taiwan; 14College of Medicine, Tzu Chi University, No. 701, Section 3, Zhongyang Rd., Hualien 970, Taiwan; 15School of Post-Baccalaureate Chinese Medicine, Tzu Chi University, No. 701, Section 3, Zhongyang Rd., Hualien 970, Taiwan

**Keywords:** nosocomial infection, ventilator circuit, ESKAPE pathogen, ventilator associated-pneumonia, full-length 16S rRNA genes

## Abstract

The rapid evolution of ventilators and their circuits, coupled with varying maximum usage durations set by different hospitals globally, poses a significant risk for the proliferation and transmission of nosocomial infections in intensive care settings. This study investigated temporal changes in bacterial community structure and predicted metabolic functions in ventilator circuits over a three-week period, with a specific focus on ESKAPE pathogens. The results of full-length 16S rRNA sequencing revealed dynamic shifts in bacterial communities, with an increased bacterial diversity and unique species prevalence in week-2 compared to week-1 and week-3. However, a marked emergence of pathogenic bacteria, including *Serratia marcescens* and *Chryseobacterium indologenes*, was observed in week-3 compared to week-1 and week-2. Additionally, the abundance of ESKAPE pathogens, including *Klebsiella pneumoniae* and *Acinetobacter baumannii*, was higher in week-3 compared to week-1 and week-2. Furthermore, the PCR analysis revealed a higher detection rate of *Pseudomonas aeruginosa* and *K. pneumoniae* in week-3 than in the previous weeks. FAPROTAX analysis further revealed a high abundance of specific functions associated with the pathogens of pneumonia, nosocomial, and septicemia in week-3 compared to the other two weeks, suggesting a shift toward more virulent or opportunistic pathogens with increased utilization of ventilator circuits. These findings highlight the microbial risks associated with prolonged use of ventilator circuits, underscoring the need for continuous microbial surveillance throughout their usage, and provide a foundation for optimizing infection control strategies in intensive care settings.

## 1. Introduction

Hospital-acquired infections (HAIs) or nosocomial infections are a significant global public health concern, particularly in intensive care units (ICUs). These infections often arise independently of the primary reason for hospitalization and can manifest even after the patient has been discharged [1]. ICU patients are especially susceptible to nosocomial infections, with a prevalence rate of up to 51% [1,2], leading to worse outcomes such as increased length of hospital stay, long-term disability, and increased mortality rate, are significantly driven by ESKAPE pathogens such as *Enterococcus* spp., *Staphylococcus aureus*, *Klebsiella* spp., *Acinetobacter* spp., *Pseudomonas aeruginosa*, and *Enterobacteriaceae*. These bacteria are notorious for causing a range of serious infections, including bloodstream infections, urinary tract infections, severe pneumonia, and post-surgical site infections [3,4,5]. The inappropriate use of antibiotics, particularly overuse, has contributed to the development of resistance mechanisms in these pathogens, complicating treatment strategies [6,7]. Additionally, the presence of persisters, a subpopulation of bacteria that can remain dormant and cause recurrent infections, further exacerbates the challenge [8]. Consequently, the global burden of HAIs is substantial, with a considerable number of patients affected, resulting in significant morbidity and mortality [9]. The National Healthcare Safety Network (NHSN) of the Centers for Disease Control and Prevention (CDC) has estimated that approximately 687,000 patients in acute care hospitals cause 72,000 deaths annually, with costs ranging between $97 and $147 billion [4], underscoring the critical need for robust surveillance and proactive infection prevention measures to mitigate the profound impact of HAIs on healthcare systems worldwide.

In ICUs, mechanical ventilators are essential for supporting patients with severe respiratory conditions. Despite their critical role, these systems can serve as potential reservoirs for nosocomial pathogens, particularly those caused by ESKAPE pathogens [10]. For instance, ventilator circuits and condensates have been identified as sites of these pathogens’ colonization, contributing to the onset of ventilator-associated pneumonia (VAP) [11,12,13], a serious complication with reported incidence rates ranging from 7% to 70% and mortality rates reaching up to 75% [13,14]. Although clinical guidelines suggest that ventilator circuits used for critically ill patients do not need to be changed daily for infection control purposes [15]. Additionally, protocols for the use of ventilator systems vary widely among hospitals globally, and the maximum safe duration for their use remains poorly characterized.

In this context, comprehensive microbial monitoring is necessary to reveal how bacterial communities evolve within ventilator systems over time. Traditional culture-based methods often miss slow-growing or unculturable organisms [16], limiting their diagnostic value. In contrast, full-length 16S rRNA sequencing has demonstrated high accuracy in identifying diverse bacterial taxa, with over 90% concordance with standard culture techniques, which remains the gold standard for diagnosing acute bacterial infections [17,18]. Therefore, this study aims to investigate temporal variations in bacterial community structure, metabolic function, and ESKAPE pathogen prevalence in ventilator circuits over a three-week ICU usage period by employing full-length 16S rRNA sequencing and PCR assay. We hypothesized that prolonged ventilator use would result in decreased microbial community structure and a shift toward more opportunistic and virulent bacterial taxa. The findings of this study could support the development of improved infection control strategies and inform evidence-based guidelines for the safe use of ventilator systems in critical care settings.

## 2. Materials and Methods

### 2.1. Sample Collection

A total of 16 ventilator circuit samples were collected from a cohort consisting of 8 male and 8 female patients, ranging in age from 34 to 94 years. The majority of these patients were suffering from respiratory illnesses and were all receiving mechanical ventilation, as shown in Appendix A. The utilization of ventilator tubes by these patients was monitored over three periods to determine the temporal changes in microbial dynamics and the risk of pathogenic bacterial population. The ventilator tubes used continuously up to day 7 were categorized as week-1, those continuously used up to day 14 and 21 were categorized as week-2 and week-3, respectively. At the end of each period, the tubes were carefully collected in sterilized specimen bags and transported to the laboratory under controlled temperature conditions. There, the ventilator tubes were rigorously rinsed with PBS buffer followed by centrifugation at 2600 rpm for 30 min. Finally, the 45 mL of supernatant was discarded, and the microbial-laden pellet was stored for subsequent DNA extraction.

### 2.2. DNA Extraction and Sequencing

The extraction of genomic DNA from ventilator tube samples was carried out using the Quick-DNA™ Fungal/Bacterial Miniprep Kit (Zymo Research, Irvine, CA, USA), strictly following the manufacturer’s protocol. The purity and quantity of the extracted gDNA were analyzed using a Nanodrop^®^ spectrophotometer (Thermo Fisher Scientific, Waltham, MA, USA) at a wavelength of 230 to 280 nm. The quality of the extracted gDNA was further assessed using gel electrophoresis (1.5% agarose gel, utilizing a Tris-acetate EDTA buffer) at a voltage of 110 V over 30 min, followed by DNA band examination under UV light. Full-length 16S rRNA sequencing targeting the V1–V9 hypervariable regions was performed using the primer set 27F (5′-AGRGTTTGATYMTGGCTCAG-3′) and 1492R (GGYTACCTTGTTACGACTT), as described previously [19]. The amplicons were sequenced according to the manufacturer’s instructions using the PacBio SMRT sequencing platform (Pacific Biosciences Inc.; San Diego, CA, USA), using a single-end method. Circular consensus sequence (CCS) reads were obtained from raw PacBio sequencing data using the standard software tools provided by the manufacturer (Pacific Biosciences). The sequencing data obtained were then analyzed using the QIIME2 (v2023.7) pipeline, where the raw reads undergo processing to trim chimeric and noisy sequences via DADA2 (q2-DADA2 v2023.7). The processed, high-quality sequence reads were clustered into amplicon sequence variants (ASVs) with a 98% similarity threshold using the NCBI reference database. Later, these ASVs were assigned taxonomic IDs, and the resulting bacterial taxonomy was collapsed (1–7) up to the species level as described previously [20]. Additionally, the variations in the bacterial diversity (alpha and beta) based on the temporal changes in the ventilator circuits were explored using Simpson, Shannon, and unweighted unifrac distance. Finally, the bacterial community functions associated with the ventilator tube were investigated using the Functional Annotation of Prokaryotic Taxa (FAPROTAX) pipeline as described previously [21,22].

### 2.3. Detection of ESKAPE Pathogens

All the ventilator tube samples were assessed to confirm the presence/absence of ESKAPE pathogens by targeting their specific genes using PCR. Intergenic spacer (ITS) PCR was employed to confirm the *A. baumannii*, whereas PCR targeting the 16S-23S rRNA gene was employed to confirm the *K. pneumoniae* in the ventilator tube based on the temporal variations. Additionally, 16S rRNA analysis was employed to confirm the presence of *P. aeruginosa* in the ventilator tube. Contrarily, *Staphylococcus aureus*, *Enterococcus faecium*, and *Enterococcus faecalis* were verified by PCR, targeting their specific genes such as *sodA* and *ddI*, respectively. The protocols for PCR amplification, including conditions and primers, are detailed in Appendix A.

## 3. Results

### 3.1. Variations in Bacterial Community Structure and Potential Health Risks Associated with Prolonged Use of Ventilator Tubes

The bacterial diversity associated with increasing durations of ventilator tube usage was explored using Shannon and Simpson alpha diversity metrics (Figure 1A,B). Both indices revealed higher bacterial diversity in week-2 compared to week-1, followed by a marked decrease in week-3. For Shannon diversity, Kruskal–Wallis analysis showed significant differences between week-1 and week-3 (*p* < 0.05) and between week-2 and week-3 (*p* < 0.05), but no significant change between week-1 and week-2. In contrast, the Simpson index showed significant differences among all three weeks (*p* < 0.05), suggesting changes in bacterial community evenness over time. Beta diversity analysis based on the unifrac distance matrix was conducted to demonstrate whether variations in bacterial community structure were attributed to individual samples or to their respective groups, namely week-1, week-2, and week-3 (Figure 1C). These results demonstrated that samples from week-1 were clustered together, indicating a similar bacterial community structure among these samples. Most of the week-2 samples, such as VC-14, VC-23, and VC-73, except VC-12 and VC-71, were clustered together with less distinct separation, indicating minor changes in bacterial community structure. However, week-3 samples showed distinct separation with less clustering, indicating significant variations in bacterial community structure and composition. These patterns were supported by PERMANOVA, which confirmed significant differences in bacterial community composition among the three weeks (*p* = 0.027).

### 3.2. Variation in Bacterial Community Composition and Potential Health Risks Associated with Prolonged Use of Ventilator Tubes

PacBio sequencing targeting the full-length 16S rRNA revealed a total of 262 classifiable species found in the samples collected from ventilator tubes utilized over the three-week period. PCA analysis was utilized to determine the variation in the bacterial community among the samples based on the duration of ventilator tube usage. In the top scatter plot (PC1 vs. PC2), all three groups overlap significantly along the PC1 axis (which explains 23.1% of the variance) but show some separation along the PC2 axis (15.5% of variance), as shown in Figure 2A. Samples from week-3, such as VC-4, VC-8, and VC-36, were slightly less clustered than those of week-1, followed by week-2, suggesting an increase in the variance of bacterial composition with prolonged use of ventilator tubes. However, this variation in bacterial community composition was more profound with the prolonged use of a ventilator compared to that with individual sample differences. This was made evident by the finding that only 2.3% of bacterial species were common across the three weeks. However, the number of unique species increased in week-2 (32.8%) compared to week-1 (30.2), but the number decreased significantly in week-3 (8.8%). Similarly, the number of common species was highest between week-1 and week-2 (15.6%), followed by week-2 and week-3 (6.9%), and lowest between week-3 and week-1 (3.4%). These results further indicate that the variation in bacterial community composition is significantly influenced by the duration of ventilator tube usage over time.

Top 10 species results showed that *Eschericatella adiacens* (50%) was the most abundant species in week-1, followed by *Corynebacterium striatum* (38%), *Streptococcus peroris ATCC 700780* (4%), and *Prevotella jejuni* (2%) (Figure 3). Contrarily, in week-2, *Serratia marcescens subsp. marcescens ATCC 13880* (37%) was the most dominant species, followed by *Stenotrophomonas pavanii* (22%), *Muribaculum intestinale* (9%), and *Chryseobacterium indologenes NBRC 14944* (7%), whereas, in week-3, *Chryseobacterium indologenes NBRC 14944* (46%) was the most dominant, followed by *Stenotrophomonas pavanii* (29%), *Pseudomonas tolaasii* (7%), and *Klebsiella pneumoniae* (7%). These results further highlight that the prolonged use of ventilator tubes leads to significant shifts in the dominant bacterial species over time, with each week showing a different profile of bacterial prevalence.

### 3.3. Variation in Pathogenic Bacterial Community Composition and Potential Health Risk Associated with the Prolonged Use of Ventilator Tube

A diverse group of pathogenic bacteria exhibited varying patterns of abundance in the ventilator tube samples over three weeks. Most pathogens were highly abundant in week-1, followed by a decline in week-2 and week-3 (Figure 4A). For instance, *Escherichia fergusonii ATCC 35469* was the most dominant pathogen followed by *Corynebacterium striatum* in week-1, but decreased in abundance in week-2 and disappeared by week-3. (Figure 4B). In contrast, *Serratia marcescens ATCC 13880* increased in abundance from week-2 to week-3, despite not being detected in week-1. Other pathogens, such as *Eggerthella lenta* and *Streptococcus agalactiae ATCC 13813,* were only found in week-2, while highly virulent pathogens like *K. pneumoniae*, *Acinetobacter baumannii*, and *Chryseobacterium indologenes NBRC 14944* were more abundant in week-3 than in the previous weeks. These results indicate a dynamic shift in pathogenic bacterial composition over time, with more virulent strains, such as *K. pneumoniae* and *Acinetobacter baumannii*, emerging as the duration of ventilator tube usage increased.

PCR analysis further confirmed an increased detection rate of ESKAPE pathogens with prolonged ventilator tube use. *P. aeruginosa* showed a marked increase, starting at 16.6% in week-1, slightly rising to 20% in week-2, and then sharply increasing to 60% in week-3. *S. aureus* maintained a stable detection rate of around 20% throughout the three weeks, while *K. pneumoniae* showed inconsistent detection, being present in week-1, absent in week-2, and reappearing in week-3 (Table 1). *A. baumannii*, *E. faecium*, and *E. faecalis* were not detected in any samples over the three-week period. These findings underscore the significant risk of nosocomial infections associated with the prolonged use of ventilator tubes.

### 3.4. Variation in Bacterial Community Metabolic Functions Associated with the Prolonged Use of Ventilator Tubes

The PCA analysis indicated significant overlap in the annotated functions of the bacterial communities over the duration of three weeks of ventilator usage (Figure 5A), although the high percentage of variance explained by PC1 (67.1%) could not clearly separate the annotated functions of the bacterial communities among the three weeks. Similarly, PC2 and PC3, which accounted for 20.1% and 7.6% of the variance, respectively, also could not provide clear separation among the three weeks. The results indicate a relatively stable microbial community function over time, with minimal temporal variations. The most abundant annotated function was chemoheterotrophy among the three weeks, with no significant difference in their relative abundance over the increasing time period of ventilator tube usage (Figure 5B). In contrast, aerobic chemoheterotrophy showed an increase in abundance over time, with a higher abundance associated with week-3, followed by week-2, than that of week-1. The functions related to human pathogens were all higher in week-1 compared to subsequent weeks. However, functions related to specific pathogens, such as pneumonia, nosocomial, and septicemia, were highly abundant in week-3 compared to the other two weeks. This pattern suggests a dynamic shift in the pathogenic bacterial community composition and their functionality over time, particularly toward an increase in virulent or opportunistic pathogens by week-3.

## 4. Discussion

The rapid changes in ventilators and their circuits in ICUs are not well-defined, posing significant risks of nosocomial infections, which lead to increased morbidity, mortality, and economic burdens [23]. This study highlights the necessity of investigating microbial communities associated with ventilator circuits to enhance patient care strategies in ICUs and reduce infection risks. The study observed an initial increase in microbial diversity between week-1 and week-2, followed by significant fluctuations in the number of unique bacterial species, with a higher count in week-2 and a notable decline by week-3. These patterns of increase and decrease in microbial populations over time could be attributed to microbial competition for survival and adaptation to the ventilator circuit environment. Previous research has also indicated that ventilator circuits are prone to significant bacterial contamination, leading to notable bacterial detection rates and concentrations [15,24].

Additionally, this study also observed a shift in the dominant bacterial species of ventilator circuits, with *E. adiacens* and *C. striatum* dominating in week-1, while *S. marcescens* and *C. indologenes* emerged as prevalent in week-2 and week-3, respectively. *S. pavanii* and *K. pneumoniae* were significantly dominant in week-3 as compared to earlier weeks. This succession might be driven by competition for resources or niche adaptation [12], as microbial communities in healthcare environments often undergo rapid shifts that allow potential pathogens to exploit weakened host defenses or disrupted microbiota. *E. adiacens* and *C. striatum* are both significant bacteria in infectious diseases. *E. adiacens* is a bacterium that has been studied in the context of host-microbe interactions, particularly using Caenorhabditis elegans as a model organism. On the other hand, *C. striatum* is a bacterium that is increasingly associated with human invasive infections, nosocomial outbreaks, and antimicrobial multidrug resistance [25]. Moreover, cases of *C. striatum* transmission from one patient to another, leading to significant hospital-acquired outbreaks, have been reported in intensive care units. Consequently, it is now recognized as an emerging pathogen in numerous countries [26]. Additionally, *S. maltophilia* primarily causes nosocomial infections in immunocompromised or debilitated patients and can adhere to moist foreign surfaces and form biofilms, making it difficult to eradicate from the hospital environment. It has been linked to a range of infections, including pneumonia, bloodstream, urinary tract, intra-abdominal, meningitis, and ocular infections; it also colonizes the airways in cystic fibrosis patients, leading to acute pulmonary exacerbations [27]. *K. pneumoniae* is a significant pathogen associated with ventilator-associated pneumonia (VAP) in intensive care units (ICUs) [28,29]. The ability of *K. pneumoniae* to form robust biofilms on abiotic surfaces and its transmission through ventilator circuits and other medical devices in ICUs contribute to its role in causing VAP [30].

PCR analysis revealed varying prevalence of ESKAPE pathogens associated with ventilator circuits over time, with *P. aeruginosa* showing an increased detection rate by week-3, highlighting the necessity for targeted infection control measures and the escalating risk of infection over time. It has been reported that *P. aeruginosa* is associated with community-acquired and nosocomial pneumonia, particularly in ventilated patients [31]. The bacterium can cause chronic infections in cystic fibrosis patients, leading to a decline in lung function and reduced survival rates [32]. Conversely, the consistent detection rate of *S. aureus* and the sporadic presence of *K. pneumoniae* suggest varying levels of threat posed by different pathogens, necessitating a nuanced approach to ventilator tube management. According to a previous study, *S. aureus* airway colonization is associated with a 15-fold greater risk of developing *S. aureus* pneumonia in critically ill patients [33]. However, it is important to note that discrepancies observed between PCR and 16S rRNA sequencing data are due to methodological differences. PCR is a highly sensitive targeted method capable of detecting specific pathogens even at low abundance [34]. In contrast, 16S rRNA sequencing provides a broader overview of the bacterial community but may underestimate low-abundance species due to primer bias and limited taxonomic resolution [35]. For example, fluctuating detection of *K. pneumoniae* by PCR compared to its variable abundance in sequencing data likely reflects these differences. Together, these methods provide a comprehensive view of pathogen prevalence and microbial community dynamics, enhancing infection risk assessment in ventilated patients.

Similarly, functions related to specific pathogens, such as pneumonia, nosocomial, and septicemia, were highly abundant in week-3 compared to the other two weeks based on FAPROTAX analysis. This suggests a shift toward metabolic functions associated with increased virulence or opportunism, further emphasizing the critical need for continuous monitoring to understand the implications of these microbial community shifts on patient health and safety. These functional trends were consistent with the taxonomic enrichment of known nosocomial pathogens such as *Klebsiella pneumoniae* and *Acinetobacter baumannii* during the same period, supporting the biological relevance of the inferred functions. However, as the predictions are based on 16S rRNA gene-based taxonomic assignments, further validation using metagenomic or phenotypic data would strengthen the functional interpretations and confirm their clinical significance.

## 5. Conclusions

This study provides crucial insights into the dynamic changes in microbial communities and their metabolic functions associated with ventilator circuits over time, emphasizing the risks of prolonged use in intensive care settings. The findings revealed that while microbial diversity initially increased, extended usage led to the emergence of more virulent pathogens, such as *Klebsiella pneumoniae* and *Acinetobacter baumannii*, particularly by the third week. There was also a notable shift in metabolic functions related to pneumonia and nosocomial infections, further indicating a rise in virulent or opportunistic pathogens over time. These results underscore the importance of continuous monitoring and timely intervention to reduce infection risks during ventilator use. However, the small sample size, combined with uneven sample distribution across time points and lack of clinical outcome data, limits the interpretation of the potential clinical impact of these microbial shifts. Therefore, future studies involving larger cohorts and integrated clinical information will be essential to determine whether these changes are associated with increased infection risk or severity, and to support effective infection control strategies.

## Figures and Tables

**Figure 1 microorganisms-13-01982-f001:**
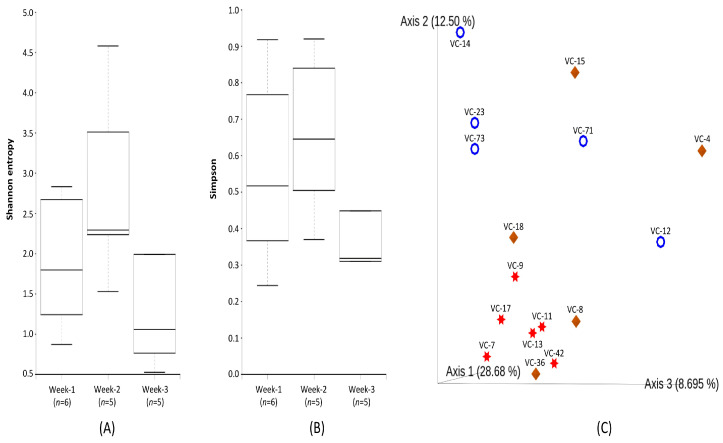
Impact of prolonged use of ventilator tubes on bacterial community structure. Alpha diversity indices (**A**,**B**) and beta diversity analysis (**C**) showing the difference in bacterial community structure among ventilator tube samples over the period of week-1, week-2, and week-3, denoted as star, ring, and diamond, respectively.

**Figure 2 microorganisms-13-01982-f002:**
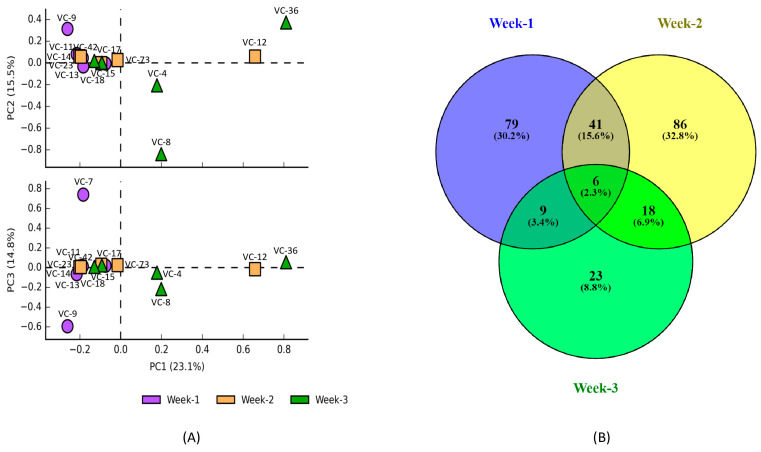
Variation in the bacterial community species composition associated with prolonged use of ventilator tube. PCA analysis (**A**) showing the variation in species composition and Venn diagram (**B**) showing the unique and shared species associated with increasing the duration of ventilator tubes across the three weeks.

**Figure 3 microorganisms-13-01982-f003:**
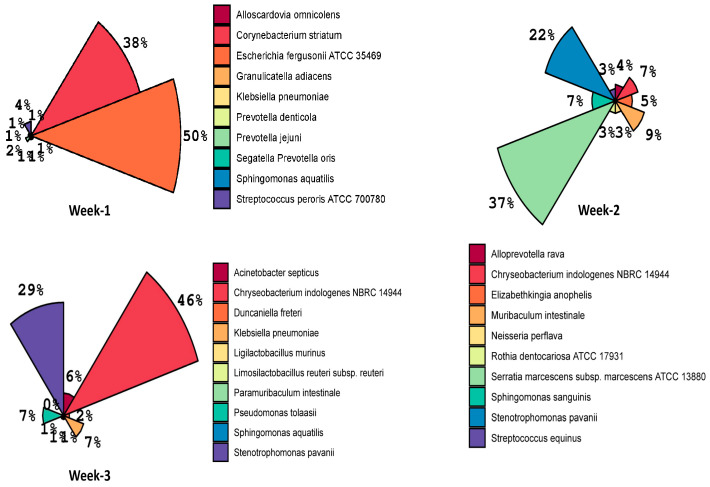
Variation in the composition and abundance of top 10 species associated with the prolonged use of ventilator tube over the period of three weeks.

**Figure 4 microorganisms-13-01982-f004:**
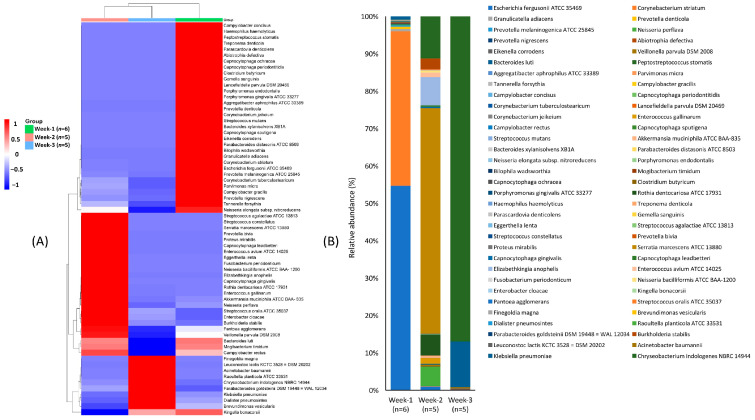
Variation in the composition and abundance of pathogenic bacteria at the species level associated with the prolonged usage of ventilator tube. Heatmap (**A**) showing the distribution pattern of pathogenic bacteria and bar plot (**B**) showing their relative abundance across the duration of ventilator tube usage over the three weeks.

**Figure 5 microorganisms-13-01982-f005:**
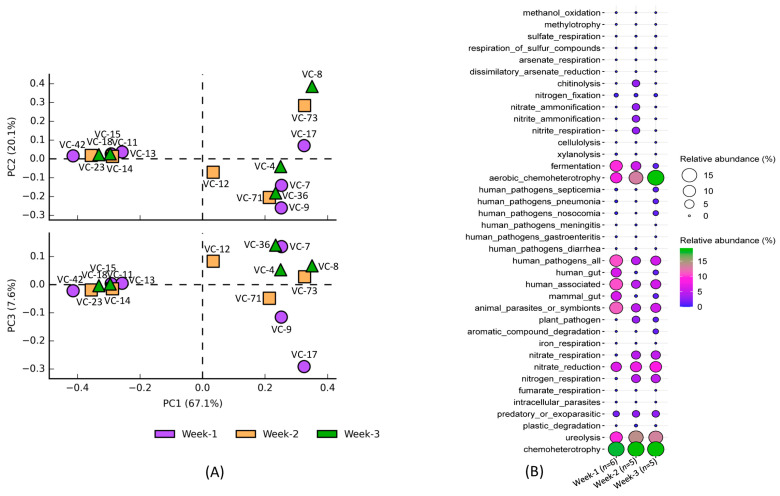
Variation in the composition and abundance of bacterial community functions associated with the prolonged usage period of ventilator tube. PCA analysis (**A**) showing the pattern of functional variation and the balloon plot (**B**) representing their relative abundance over the period of three weeks.

**Table 1 microorganisms-13-01982-t001:** Prevalence of ESKAPE pathogens associated with the prolonged use of ventilator tubes using PCR.

Pathogen	Week-1 (*n* = 6)	Week-2 (*n* = 5)	Week-3 (*n* = 5)
*P. aeruginosa*	1/6 = 16.6%	1/5 = 20%	3/5 = 60%
*A. baumannii*	0/6 = 0%	0/6 = 0%	0/6 = 0%
*S. aureus*	1/6 = 16.6%	1/5 = 20%	1/5 = 20%
*E. faecium*	0/6 = 0%	0/6 = 0%	0/6 = 0%
*E. faecalis*	0/6 = 0%	0/6 = 0%	0/6 = 0%
*K. pneumoniae*	1/6 = 16.6%	0/6 = 0%	1/5 = 20%

## Data Availability

The original contributions presented in this study are included in the article/Appendix A. Further inquiries can be directed to the corresponding authors.

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
