# Peer review of "Deciphering Bacterial Community Succession and Pathogen Dynamics in ICU Ventilator Circuits Through Full-Length 16S rRNA Sequencing for Mitigating the Risk of Nosocomial Infections"

_microorganisms, 2025, doi:10.3390/microorganisms13091982_

Round 1

Reviewer 1 Report

Comments and Suggestions for Authors

Thank you for the opportunity to review the manuscript titled "Deciphering Temporal Variations of Bacterial Community, Functionality, and Pathogen Distribution Associated with Ventilator Circuits for Mitigating and Strategizing the Risk of Nosocomial Infections." The topic is certainly relevant and the study is interesting; however, I believe some minor revisions are necessary before the manuscript can be considered for publication.

First, the title appears too broad and does not clearly reflect the specific objectives or scope of the study. As currently written, it gives the impression of a review article rather than an original research paper. A more concise and targeted title that clearly communicates what was investigated would improve the manuscript significantly.

In addition, there are some inconsistencies in terminology throughout the manuscript. For example, the acronym ESKAPE is sometimes written as "ESKAP" or otherwise inconsistently. Please ensure that "ESKAPE" is used consistently and correctly throughout the manuscript.

There are also numerous instances where scientific names are not italicized, which should be corrected to conform with standard scientific writing conventions.

Regarding the conclusions, although they are generally supported by the data, I suggest adding a more explicit discussion on the limitations of the study and possible next steps for future research. It is important to clarify that while changes in the bacterial community are noteworthy, they do not necessarily imply an increased risk or severity of nosocomial infections. This distinction should be made clear to avoid overinterpretation of the findings.

I hope these comments are helpful in strengthening the manuscript.

Author Response

We sincerely thank the reviewer for their insightful and constructive comments, which have significantly improved the clarity and scientific quality of our manuscript. We have carefully revised the text to address the majority of the suggestions provided.

Reviewer 2 Report

Comments and Suggestions for Authors

This study examines temporal shifts in bacterial communities and metabolic functions within ventilator circuits over three weeks, with a focus on the emergence of ESKAPE pathogens and their implications for nosocomial infection control. Although the paper presents a certain scientific interest, there are some concerns regarding the validity and the overall results.

  1. The manuscript presents observational data but does not clearly define a central hypothesis or specific research questions. Please revise the introduction to articulate a testable hypothesis or define the key questions addressed by the study.
  2. Sixteen samples over three weeks with unclear distribution per time point are insufficient to draw generalized conclusions. Please provide statistical power calculations or justification for the sample size to support robustness and reproducibility.
  3. The manuscript omits critical clinical parameters (e.g., patient diagnosis, duration of ventilation, antibiotic usage, and comorbidities) that may influence microbial communities. These are necessary for interpreting shifts in bacterial populations. Include or discuss these confounders.
  4. FAPROTAX predictions are based on taxonomic inference, not functional genomics. Claims regarding virulence or pathogenicity based solely on FAPROTAX need to be tempered. Please clarify these limitations and avoid overinterpretation of inferred functions.
  5. PCR detection of certain pathogens (e.g., pneumoniae) reveals fluctuations that are not reconciled with 16S rRNA sequencing data. Clarify methodological differences in sensitivity and specificity, and address any potential discrepancies between methods.
  6. There is limited mention of statistical testing (e.g., for diversity indices, PCA separations). Include appropriate statistical significance values and indicate whether ANOVA, PERMANOVA, or similar tests were applied.
  7. The manuscript inconsistently uses "ESKAP" vs "ESKAPE" pathogens. Please use standard terminology ("ESKAPE") throughout.
  8. While the study reveals microbial shifts, the implications for infection control policy, ventilator replacement intervals, or prophylactic strategies are not fully explored. Enhance the conclusion with specific clinical or procedural recommendations based on the data.

Author Response

(The authors gave the same response as above.)

Reviewer 3 Report

Comments and Suggestions for Authors

This manuscript presents a well-conceived study on the temporal dynamics of bacterial communities and pathogen profiles in ventilator circuits using full-length 16S rRNA sequencing. The research is of clinical relevance, especially in the context of nosocomial infection control in ICU settings. The integration of both taxonomic and functional microbial analysis provides valuable insights into potential risks associated with prolonged ventilator use.

However, while the scientific content is solid, the manuscript would benefit significantly from revisions in language and formatting to meet the standards of international publication. Below are my detailed suggestions.

1] Enhance the clarity of Figures 1, 4 and 5.

2] Line 62: correct the phrase " are especially are particularly,”.

3] Multiple references in the manuscript are incorrectly formatted, with redundant brackets (e.g., “[1]]”, “[6,7]]”).

4] Consider rephrase: Line 84: "remains largely undetermined".

5] Improve transition sentences between sections to ensure smoother flow.

6] “ESKAPE” sometimes appears as “ESKAP” (missing 'E').

7] Several DOI in references are missing.

8] Consider revise abstract and conclusions.

Author Response

(The authors gave the same response as above.)
